# Maternal Exposure to Particulate Matter during Pregnancy and Adverse Birth Outcomes in the Republic of Korea

**DOI:** 10.3390/ijerph16040633

**Published:** 2019-02-21

**Authors:** Yu Jin Kim, In Gyu Song, Kyoung-Nam Kim, Min Sun Kim, Sung-Hoon Chung, Yong-Sung Choi, Chong-Woo Bae

**Affiliations:** 1Department of Pediatrics, Kyung Hee University School of Medicine, Seoul 02447, Korea; yjyj42@gmail.com (Y.J.K.); feelhope@gmail.com (Y.-S.C.); 2Central Hospice Center, National Cancer Center, Goyang-si, Gyeonggi-do 10408, Korea; 3Division of Public Health and Preventive Medicine, Seoul National University Hospital, Seoul 03080, Korea; kkn002@snu.ac.kr; 4Department of Preventive Medicine, Seoul National University College of Medicine, Seoul 03080, Korea; 5Department of Pediatrics, Seoul National University Hospital, Seoul 03080, Korea; singrumi@gmail.com; 6Department of Pediatrics, Kyung Hee University Hospital at Gangdong, Kyung Hee University School of Medicine, Seoul 05278, Korea; pedc@khnmc.or.kr (S.-H.C.); baecw@khnmc.or.kr (C.-W.B.)

**Keywords:** maternal exposure, particulate matter, preterm birth

## Abstract

Air pollution has become a global concern due to its association with numerous health effects. We aimed to assess associations between birth outcomes in Korea, such as preterm births and birth weight in term infants, and particulate matter < 10 µm (PM_10_). Records from 1,742,183 single births in 2010–2013 were evaluated. Mean PM_10_ concentrations during pregnancy were calculated and matched to birth data by registered regions. We analyzed the frequency of birth outcomes between groups using WHO criteria for PM_10_ concentrations with effect sizes estimated using multivariate logistic regression. Women exposed to PM_10_ > 70 µg/m^3^ during pregnancy had a higher rate of preterm births than women exposed to PM_10_ ≤ 70 µg/m^3^ (7.4% vs. 4.7%, *P* < 0.001; adjusted odds ratio (aOR) 1.570; 95% confidence interval (CI): 1.487–1.656). The rate of low birth weight in term infants increased when women were exposed to PM_10_ > 70 µg/m^3^ (1.9% vs. 1.7%, *P* = 0.278), but this difference was not statistically significant (aOR 1.060, 95% CI: 0.953–1.178). In conclusion, PM_10_ exposure > 70 µg/m^3^ was associated with preterm births. Further studies are needed to explore the pathophysiologic mechanisms and guide policy development to prevent future adverse effects on birth outcomes.

## 1. Introduction

Air pollution in the Republic of Korea (Korea) has been an issue for many years. In particular, the particulate matter (PM) component of air pollution is becoming increasingly important due to several factors, including geographical characteristics, the chemical evolution in Seoul, and overcrowding in urban areas. In 2016, the United States National Aeronautics and Space Administration, the Ministry of Environment in Korea, and the National Institute of Environmental Research studied PM in Korea and released the Korea–United States Air Quality Study, reporting that the PM emitted domestically may exceed the recommendations outlined in the World Health Organization (WHO) air quality guidelines [1]. In previous studies, PM such as PM_2.5_ (fine inhalable particles with diameters that are generally smaller than 2.5 µm) and PM_10_ (inhalable particles with diameters that are generally smaller than 10 µm) have been associated with increased mortality and morbidity from multiple health conditions, including cardiovascular disease, lung cancer, acute respiratory infections, asthma, and diabetes [2,3,4,5,6,7,8,9]. Moreover, other studies have reported that maternal exposure to PM during pregnancy may increase the risk of preterm birth (gestational age < 37 weeks) [10,11,12]; low birth weight (birth weight < 2500 g) in term infants [13,14,15]; and congenital malformation [16] through processes related to inflammation, oxidative stress, endocrine disruption, and impaired oxygen transport across the placenta [17]. In addition, there have been several studies reporting on the long-term effects of prenatal air pollution exposure on neurodevelopment and respiratory outcomes [18,19].

Preterm birth has short-term effects on respiratory, central nervous system, and cardiovascular functions in the form of patent ductus arteriosus, respiratory distress syndrome, and intravascular hemorrhage [20]. There are also, long-term consequences for physical health, neurodevelopment, pulmonary function, and adult health (cerebral palsy, asthma, growth impairment, and hypertension) [21]. Preterm birth is the second-most common cause of mortality in children under five years of age, and low-birth weight infants (LBWIs) have a 20-fold higher mortality rate than infants with a birth weight > 2500 g [22,23].

Given the increasing attention on the relationship between exposure to PM and adverse birth outcomes, this study aimed to assess the association of birth outcomes, such as preterm births and low birth weight in term infants, with PM in ambient air pollutants in Korea. We used a cutoff value for PM concentration of 70 μg/m^3^ to indicate high levels of exposure. This reference value of 70 µg/m^3^ was based on the interim target-1 in the WHO air quality guidelines [1] and is associated with a 15% higher long-term mortality risk relative to the guideline level of 20 µg/m^3^. We hypothesized that air pollution would be negatively associated with birth outcomes, particularly preterm births and low birth weight.

## 2. Materials and Methods

### 2.1. Study Design and Participants

This nationwide registry-based study analyzed data from 1,862,441 live births in 2010–2013 registered in the Korean national birth registry. The birth weight of term infants and the proportions of preterm births and low birth weight were analyzed according to the concentration of PM_10_ during pregnancy. In Korea, all parents must report their child’s birth within 1 month and include the following information: month and date of birth; maternal residential address at the time of birth; place of birth (hospital or elsewhere); parental ages; gestational age (weeks and days); sex; birth weight; birth order; total number of births; and parental education, occupation, and nationality. All data were acquired from Statistics Korea [24], which offers data (except for date of birth and days of gestational age, which are withheld for reasons of privacy protection) to all researchers who submit the objective of a study to their website. Because the individual identifier number was removed from the individual record to protect the privacy of the individuals, each birth record was treated as a separate family member, even though a couple may have had more than one child during the study period. Since multiple births are a major factor associated with preterm birth and low birth weight, 59,516 records (3.2%) involving multiple births were excluded. In addition, we excluded 1299 records with a gestational age < 23 weeks or a birth weight of <400 g according to the guideline for withholding of neonatal resuscitation [25]. Moreover, to compare premature births with normal births, we excluded 59,443 observations with a gestational age of ≥42 weeks or a birth weight > 4000 g (Figure 1). Thus, this study evaluated data from 1,742,183 records.

### 2.2. Air Monitoring Data

To assess PM_10_ exposure, we obtained daily mean levels from the National Health Insurance Service (NHIS), which makes air pollution data available through its data sharing website [26]. The level of PM_10_ was measured using the ß-ray absorption method at 266 monitoring stations scattered throughout Korea during 2009–2013 [27]. Residential addresses (city, county, district; or si, gun, gu in Korean) at birth were utilized for spatial exposure assessment and assignment of PM_10_ concentrations during pregnancy were assigned according to the address; the study population was postulated to be stable (did not change) over the exposure time period. We calculated monthly averages of PM_10_ concentrations for each address and matched them to individuals based on their gestational age.

### 2.3. Statistical Analysis

Using ArcGIS Maps for Office (ESRI Inc., Redlands, CA, USA), we visualized PM_10_ concentrations and the proportion of preterm births in Korea according to addresses (city, county, district; or si, gun, gu) of monitoring stations and residences. We performed multiple linear regression analysis to assess correlations between mean PM_10_ level and birth weight in term infants. Multivariable logistic regression was used to assess the effect of PM_10_ in each residential region on birth weight in term infants and preterm births. The mean PM_10_ concentrations were categorized in two ways. First, we compared birth outcomes of subjects in the lower first to third quartiles with the subjects in the fourth quartile. As the 75th percentile of the concentration was 54.5325 µg/m^3^, we calculated odds ratios (ORs) using ≤54.5325 µg/m^3^ as the reference category. Second, proportions and ORs were analyzed based on ≤70 µg/m^3^, which is the interim target-1 of the WHO [1]. To evaluate the difference between metropolitan areas (Seoul, Busan, Daegu, Incheon, Gwangju, Daejeon, and Ulsan) and nonmetropolitan regions, we performed sensitivity analyses and assessed differences in ORs among both regional groups. We estimated adjusted ORs (aORs) after controlling for variables known to affect birth outcomes [28,29,30], including season at birth, parity, and parental job, education level, age, nationality, and residential region (capital region or not). 

### 2.4. Ethics Statement

This study was granted ethical exemption by the institutional review board at Kyung Hee University Hospital (Seoul, Korea), since this was a secondary analysis of de-identified data (IRB No. 2018-01-091). The study was conducted in accordance with the Declaration of Helsinki.

## 3. Results

### 3.1. Demographic and Birth-Related Characteristics

Table 1 presents the baseline characteristics of the analyzed birth records. Overall, the mean ± standard deviation of gestational age and birth weight were 38.7 ± 1.5 weeks and 3200 ± 400 g, respectively. Preterm infants accounted for 4.7% of neonates and LBWIs accounted for 3.8%. Rates of preterm births and low birth weight increased over the observation period. Preterm births accounted for 4.6% of births in 2010 and increased to 4.9% in 2013. Proportions of low birth weight among term infants (28,728/1,659,659 × 100) were constant at 1.7% over the observation period.

### 3.2. Distribution of PM_10_ and Preterm Births in Korea

Median concentration of PM_10_ over five years decreased from 46 to 43 µg/m^3^, with the lowest level reported in 2012 (Table 2). Figure 2a illustrates concentrations of PM_10_ in each region divided into quarters. Concentrations were high in the capital area and metropolitan areas. In the capital area, the quality of air was worse in the west coast areas and rural areas that have factories than it was in Seoul, the capital city. Figure 2b presents proportions of preterm birth by region. Preterm births occurred more frequently in the west coast areas in the capital region and noncapital regions than in Seoul.

### 3.3. PM_10_ and Birth Outcomes

A 10 µg/m^3^ increase in concentration of PM_10_ during pregnancy was associated with a 1 g decrease in birth weight among term infants (*P* = 0.001). The proportion of low birth weight in term infants was higher when the exposure to mean PM_10_ concentration was >70 µg/m^3^ than it was when exposure was ≤70 µg/m^3^ (1.9% vs. 1.7%), although the aOR was not significantly higher than the reference group (aOR: 1.060, 95% confidence interval (CI): 0.953–1.178, *P* = 0.283) (Table 3). Women in the highest quartile had higher odds of preterm birth compared with women in the lower three quartiles (54.5325 µg/m^3^ or less) of PM_10_ exposure (aOR: 1.044, 95% CI: 1.025–1.062, *P* < 0.001). In particular, women exposed to PM_10_ >70 µg/m^3^ during pregnancy had a significantly higher proportion of preterm births (7.4% vs. 4.7%) than those exposed to ≤70 µg/m^3^ (aOR: 1.570, 95% CI: 1.487–1.656, *P* < 0.001). The aOR for the relationship between exposure to PM_10_ >70 µg/m^3^ for very preterm births (gestational age less than 32 weeks) was also statistically significant (aOR: 1.966, 95% CI: 1.776–2.177, *P* < 0.001) (Table 3).

In metropolitan areas, preterm births were more prevalent among those exposed to the highest quartile of PM_10_ than those exposed to lower PM_10_ levels (5.4% vs. 4.6%), and the aOR was statistically significant (aOR: 1.156, 95% CI: 1.123–1.190, *P* < 0.001). However, there was no difference between groups in nonmetropolitan regions. aORs for the association between preterm births and mean PM_10_ exposure > 70 µg/m^3^, were statistically significant, irrespective of region (Table 4).

## 4. Discussion

Using a nationwide registry-based study, we analyzed the associations of between preterm births and birth weight of term infants and exposure concentrations of PM_10_ during pregnancy. 

Our results indicate that exposure to PM_10_ > 70 µg/m^3^ during pregnancy may be associated with preterm births. There is an ongoing debate regarding the health effects of exposure to PM_10_ during pregnancy, but our results are consistent with most other studies [11,12]. Our results regarding the association of higher PM_10_ concentrations and increased preterm birthrate closely agree with several previous studies on the effects of PM_10_ exposure on preterm birth in a restricted area in Korea [31,32]. However, unlike previous research, this study analyzed not only the distribution of PM but also the specific standard of 70 μg/m^3^, which is associated with a higher long-term mortality risk according to the WHO [1]. When the results are visualized geographically, it is clear that preterm births occurred more frequently in west coast areas, and some rural cities in the capital area where there are many more factories than in Seoul. Moreover, exposure to PM_10_ > 70 µg/m^3^ during pregnancy was significantly associated with very preterm births. In addition, mean exposure to PM_10_ > 70 µg/m^3^ resulted in significantly elevated adjusted odds ratios regardless of whether the location was a metropolitan or nonmetropolitan area. 

Finally, we found a tendency for increases in PM_10_ to be associated with increased risk of adverse birth outcomes such as low birth weight in term infants. Our data showed a statistically significant (*P* = 0.001) 1 g decrease in birth weight among term infants per 10 µg/m^3^ increase in PM_10_ exposure during pregnancy. This tendency is consistent with previous research, but the changes in birth weight described previously have been so modest that they may have little clinical importance [33,34]. The proportion of low birth weight in term infants was higher in areas where the mean PM_10_ concentration was >70 µg/m^3^, but the association was not statistically significant, as measured by the aOR. 

Although NHIS data showed a decline in PM_10_ concentration over a five-year period, ambient air pollutants—specifically particulate matter—remain a burden on the economy and human society in Korea and worldwide. The Global Burden of Disease, Injuries, and Risk Factors Study of 2016 identified air pollution, especially ambient air pollution, as the sixth leading risk factor for global disease [35]. At the same time, according to 2016 data from the Korean National Statistical Office (KNSO), the frequency of preterm births at a gestational age of <37 weeks increased by 1.5 times from the level seen 10 years previously, to 7.2%. The prevalence of LBWIs (under 2500 g at birth) increased 0.2% from the 2015 level, and steadily increased to about twice the level seen in 1996. Meanwhile, preterm birth is the second-most common cause of mortality in children under 5 years of age, and LBWIs have a 20-fold higher mortality rate than infants of normal birth weight (>2500 g). Therefore, policies are needed to ameliorate modifiable factors for adverse birth outcomes, such as air pollution, to reduce the preterm birthrate and the rate of LBWIs. A campaign is needed to educate the public, especially pregnant women and their families, on methods to reduce or avoid PM. 

Our study had several strengths, including adjustment for a number of covariates, including maternal age, parity, infant sex, and parental employment, which have been associated with adverse birth outcomes in previous reports [28,29,30]. In addition, our analysis was based on the specific criteria of 70 µg/m^3^ of PM_10_, which is relevant to higher risks. However, it is important to note some limitations. First, when we discuss the association between PM and adverse birth effects, such as preterm birth and low birth weight, we should consider other factors that may affect birth results. These include the family history of preterm births, maternal smoking history, low maternal body mass index, prior preterm birth, medical and pregnancy history, occupational exposure, and other factors. These factors could not be analyzed due to the unavailability of the necessary data. Second, until recently, the concentrations of PM_10_ and PM_2.5_, which is also thought to be related to adverse birth outcomes, had not been measured in Korea [13,14,36,37]. Because there are various types of air pollutants, including SO_2_, O_3_, and NO_2_, comprehensive analyses of the effects of pollutants beyond PM_10_ on birth outcomes are warranted. Lastly, this study did not analyze results by trimester because detailed data regarding birth history were not available from the KNSO. There was also no data regarding the date of birth or the day of gestational age; therefore, there was no possibility of dividing pregnancies into trimesters.

In April 2017, The Lancet published The Lancet Planetary Health to assess the effects of environment change on human health, but also to investigate other factors such as political, economic, and social systems that govern those effects [38]. This reflects the increasing global emphasis on the importance of the environment on human health.

## 5. Conclusions

In conclusion, we found that exposure to ambient air pollutants during pregnancy, especially PM_10_, was associated with an increased rate of preterm births. Future research priorities should include explorations of the pathophysiological mechanism behind this association. Long-term, multifaceted studies are also needed to guide development of policies to prevent the adverse effects of air pollutants on birth outcomes in the future.

## Figures and Tables

**Figure 1 ijerph-16-00633-f001:**
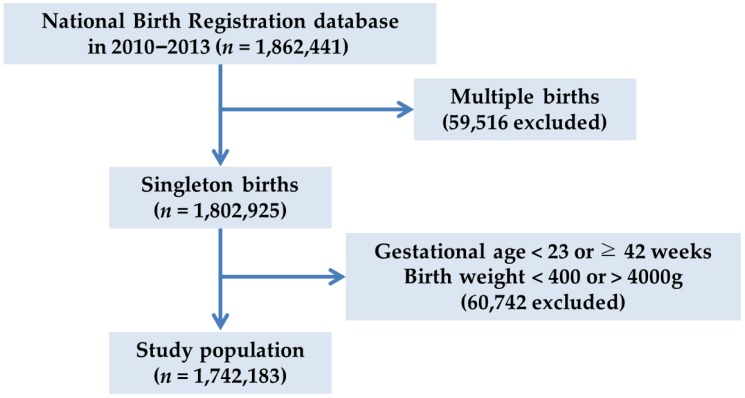
Flow diagram of the study population.

**Figure 2 ijerph-16-00633-f002:**
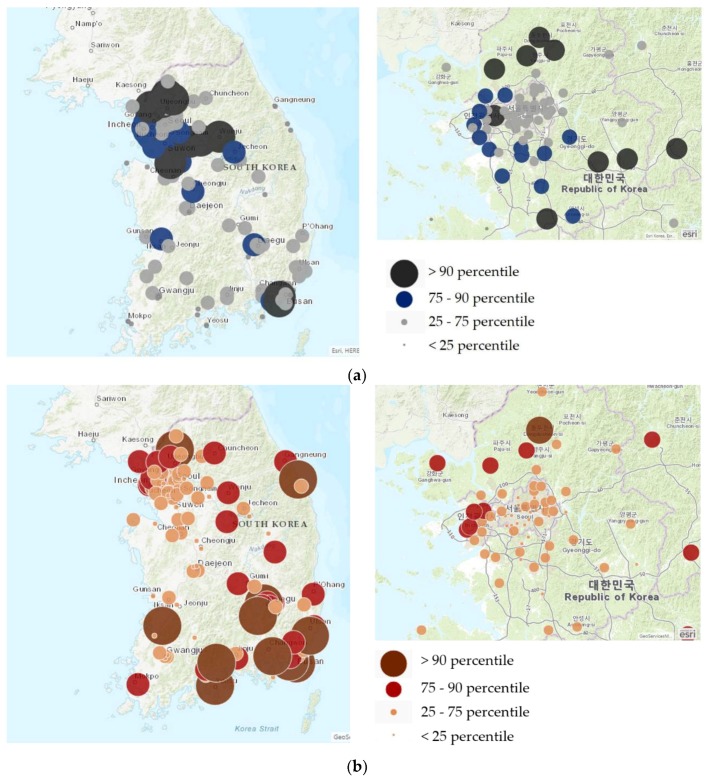
(**a**) Distribution of particulate matter less than 10 μm in Korea. (**b**) Distribution of preterm births in Korea.

**Table 1 ijerph-16-00633-t001:** Background characteristics of births between 2010 and 2013 in Korea.

Characteristics	*n* (%) or Mean (SD) (*n* = 1,742,183)
**Infant sex**	
Male	888,711 (51.0)
Female	853,472 (49.0)
**Marital status**	
Married	1,703,795 (97.9)
Unmarried	36,857 (2.1)
**Paternal age (years)**	
Mean	33.7 (4.6)
<20	2752 (0.2)
20–29	268,567 (15.6)
30–39	1,282,258 (74.3)
≥40	172,350 (10.0)
**Maternal age (years)**	
Mean	31.0 (4.1)
<20	10,707 (0.6)
20–34	1,414,366 (81.2)
≥35	316,175 (18.2)
**Area of birth**	
Capital region	885,157 (50.8)
Others	857,026 (49.2)
**Place of birth**	
Hospital	1,713,436 (98.4)
Others	27,627 (1.6)
**Paternal education**	
University or higher	1,227,626 (71.2)
High school or lower	495,918 (28.8)
**Maternal education**	
University or higher	1,215,714 (70.0)
High school or lower	521,532 (28.0)
**Paternal employment**	
Manager or specialist	477,105 (27.4)
Officer	585,437 (33.6)
Service	296,117 (17.0)
Blue collar	317,360 (18.2)
Unemployed ^a^	66,164 (3.8)
**Maternal employment**	
Manager or specialist	221,519 (12.7)
Officer	245,619 (14.1)
Service	78,395 (4.5)
Blue	34,344 (2.0)
Unemployed ^a^	1,162,306 (66.7)
**Paternal nationality**	
Korean	1,715,981 (99.4)
Non-Korean	11,048 (0.6)
**Maternal nationality**	
Korean	1,679,145 (96.5)
Non-Korean	60,629 (3.5)
**Parity**	
Primiparous	899,141 (51.7)
Multiparous	840,365 (48.3)
**Mean gestational age (weeks)**	38.7 (1.5)
**Preterm infants**	82,524 (4.7)
**Mean birthweight (kg)**	3.2 (0.4)
**Low birthweight in term infants**	28,728 (1.7 ^b^)

^a^ Unemployed: unemployed, housewife, or student. ^b^ Proportion of low birthweight in term infants: 28,728/1,659,659 × 100.

**Table 2 ijerph-16-00633-t002:** Distribution of particulate matter less than 10 µm (PM_10_) between 2009 and 2013 in Korea.

PM_10_ (µg/m^3^)	2009	2010	2011	2012	2013	Total
1st centile	12	10	9	10	13	11
25th centile	33	30	30	28	30	30
Median	46	45	44	40	43	44
Mean	53	51	50	45	49	50
75th centile	65	64	62	56	60	62
90th centile	91	88	84	76	82	84
99th centile	167	170	179	118	139	156
Range	155	160	170	108	126	145

**Table 3 ijerph-16-00633-t003:** Associations of low birth weight in term infants and preterm births with quartiles of particulate matter less than 10 µm (PM_10_) and the interim target-1 of WHO (≤70 µg/m^3^).

	Exposure	Proportion (%)	*P*-Value	Adjusted OR (95% CI)
Low birthweights in term infants	1st–3rd	1.7		
	4th	1.8	0.495	1.010 (0.981–1.040)
	≤70 µg/m^3^	1.7		
	>70 µg/m^3^	1.9	0.283	1.060 (0.953–1.177)
Preterm infants	1st–3rd	4.7		
	4th	4.9	<0.001	1.044 (1.025–1.062)
	≤70 µg/m^3^	4.7		
	>70 µg/m^3^	7.4	<0.001	1.570 (1.487–1.656)
Very preterm infants (Gestational age < 32 weeks)	1st–3rd	1.0		
	4th	1.1	<0.001	1.095 (1.055–1.137)
	≤70 µg/m^3^	1.0		
	>70 µg/m^3^	2.0	<0.001	1.966 (1.776–2.177)

OR adjusted for parity, parental job, education level, age, nationality, residential regions (capital region or not), and season at birth.

**Table 4 ijerph-16-00633-t004:** Associations of preterm births and maternal exposure to particulate matter less than 10 µm (PM_10_) during pregnancy according to the residential region.

	Exposure	Proportion (%)	*P*-Value	Adjusted OR (95% CI)
Metropolitan areas	1st–3rd	4.6		
	4th	5.4	<0.001	1.156 (1.123–1.190)
	≤70 µg/m^3^	4.7		
	>70 µg/m^3^	8.9	<0.001	1.934 (1.666–2.247)
Non-metropolitan regions	1st–3rd	4.7		
	4th	4.7	0.127	0.984 (0.963–1.006)
	≤70 µg/m^3^	4.7		
	>70 µg/m^3^	7.2	<0.001	1.521 (1.436–1.611)

OR adjusted for parity, parental job, education level, age, nationality, and season at birth.

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
