# Peer review of "Maternal Exposure to Particulate Matter during Pregnancy and Adverse Birth Outcomes in the Republic of Korea"

_ijerph, 2019, doi:10.3390/ijerph16040633_

Round 1

Reviewer 1 Report

1)      The first two sentences from the abstract can be combined into one.

2)      In line 16 ”was significantly high” can be deleted

3)      The conclusion from the study “We suggest that pregnant women avoid long-term exposure to that level of air pollutants” is strange. How individual women can avoid that exposure (not going outdoors?). I would rather say that legislation or improvement of  air quality need to be strengthened.

4)      Neurodevelopmental and respiratory outcomes as the long-term consequences of prenatal air pollution exposure should be also mentioned (line 47-50).

5)      Under the methods sections the authors have indicated: “In Korea, all parents must report their child’s birth within 1 month…..” so if it is based on parental report how accurate are the data (can the bias operate?)

6)      What does it mean “the study population was postulated to be stable (did not move) over the exposure time period….” – is it only suspected or proven?

7)      It is not clear from the description that PM10 was calculated for the whole pregnancy period or only for the month of gestational age (which in my opinion

8)      The conclusions from the study need to be improved

Author Response

Reviewer 1

Comment #1: The first two sentences from the abstract can be combined into one.

Thank you for your suggestion. We have combined the first two sentences into one (page 1, line 17).

“Air pollution has become a major global health concern due to its association, associated with numerous health effects“

Comment #2: In line 16 “was significantly high” can be deleted.

We deleted “significantly high” and modified the sentence (page 1, lines 25-27).

“Women exposed to PM10 >70 µg/m3 during pregnancy had a higher rate of preterm births than women exposed to PM10 ≤70 µg/m3 (7.4% vs. 4.7%, P <0.001; adjusted odds ratio (aOR) = 1.570; 95% confidence interval (CI): 1.487–1.656).”

Comment #3: The conclusion from the study “We suggest that pregnant women avoid long-term exposure to that level of air pollutants” is strange. How individual women can avoid that exposure (not going outdoors?). I would rather say that legislation or improvement of air quality need to be strengthened.

Comment #8: The conclusions from the study need to be improved.

Thank you for your important suggestions. We attempted to emphasize both approaches (individual efforts and policy-making) for preventing exposure of pregnant women to PM10, but we understand how our advice could be understood as impractical. Based on your suggestion, we have modified the conclusions (page 1, line 31-32 & page 9, lines 245-250). Thank you again, we feel that these revisions represent an important improvement to the manuscript.

“Further studies are needed to explore the pathophysiologic mechanisms and guide policy development to prevent future adverse effects on birth outcomes”

“In conclusion, we found that exposure to ambient air pollutants during pregnancy, especially PM10, was associated with an increased rate of preterm births. Future research priorities should include explorations of the pathophysiological mechanism behind this association. Long-term, multifaceted studies are also needed to guide development of policies to prevent the adverse effects of air pollutants on birth outcomes in the future.”

Comment #4: Neurodevelopmental and respiratory outcomes as the long-term consequences of prenatal air pollution exposure should be also mentioned (line 47-50).

We added the long-term consequences and references (page 2, lines 51-53)

“Also, there have been several studies reporting on the long-term effects of prenatal air pollution exposure on neurodevelopment and respiratory outcomes [18, 19].”

Comment #5: Under the methods sections the authors have indicated: “In Korea, all parents must report their child’s birth within 1 month…..” so if it is based on parental report how accurate are the data (can the bias operate?)

In Korea, parents report birth information of their children at government offices. They visit the office with the medical certificate which is written by the obstetrician and report the information under the supervision of civil servants, making this information quite reliable.

Comment #6: What does it mean “the study population was postulated to be stable (did not move) over the exposure time period….” – is it only suspected or proven?

Thank you for your question. As we used birth registry data, we could not track migration of individual subjects in the population during pregnancy. Therefore, we relied on the assumption that women stayed in the same region during pregnancy.

Comment #7: It is not clear from the description that PM10 was calculated for the whole pregnancy period or only for the month of gestational age (which in my opinion

Thank you for your important question, and we regret that we cannot find your full comment on the journal’s website. As we reported in the methods and the discussion, the birth registry provided only the month of birth and week of gestational age. Therefore, we used the average PM10 level for the entire pregnancy period.

Reviewer 2 Report

This manuscript explored associations between PM10 exposure during gestation and adverse birth outcomes, including preterm birth, low birth weight, and birth weight of term infants. While many current literatures focus on the relationship between adverse birth outcomes and fine particulate matter (PM2.5), this article could not investigate this relationship due to unavailability of data. The unique part of this article is that the authors used WHO interim target-1 value of PM10 (70µg/m3), and compared effect estimates between attained vs non-attained subjects. The authors found strong associations in most models.

Overall it is clearly presented and easy to follow. However, I have several concerns for exposure definition and modeling.

1.      It is not clear the way PM10 exposure was assigned to each mother. It said “We calculated monthly averages of PM10 concentrations for each address and matched it to individuals based on their gestational age.” Many previous published papers used weekly average rather than monthly to calculate gestational exposure. This will avoid exposure misclassification. If it was assigned monthly average, someone who delivered the baby on the 1st day of the month and someone who delivered the baby on the 30th in the same month will be assigned same pollution exposures as their last month of exposure. Exposure misclassification is somewhat concern; approximately 4 weeks difference of exposure period.

2.      Regarding the comment above, how much is it reliable on gestational age? Are they determined based on self-report or ultrasound? In determining gestational exposure, how did you calculate conception date?

3.      In the limitation paragraph of the discussion, the authors wrote “there was also no data regarding the date of birth,” but the method section said date of birth is available (line 71). Please clarify this.

4.      In the statistical model, the authors put infants’ sex as an adjustment variable. This should be taken out from the model. It is true that infants’ sex is strongly tied with adverse birth outcomes. Nevertheless, this variable is not associated with gestational exposure to PM10. Unless the variable is associated with exposure variable, the variable does not satisfy the requirement as an adjustment variable in the epidemiological model. Please refer to the epidemiological textbook to ensure this point.

5.      The authors evaluated two different type of birth outcomes. One is fetal growth, which is evaluated by birth weight, and the other is gestational length, which is evaluated by gestational age. In evaluating gestational length (i.e. preterm birth), exposure period should be adjusted to the relevant exposure period. Even if the exposure length is 40 weeks for a given infant, their exposure period should be adjusted to 37 weeks since exposure after 37 weeks are non-relevant to the outcome, preterm birth. This is same for very preterm birth. Any birth after 32weeks should assign pollutant level until 32nd gestational weeks.

6.      In preterm analysis, the authors used birth weight as a confounder. Again, I don’t think this variable is qualified as a confounder in the epidemiological model.

7.      Low birth weight and preterm birth is including infants delivered before 37 weeks, but birth weight is evaluated only for term infants. The number of term birth should be mentioned in the text as well as table 1.

8.      In the analysis, exposure was compared between low three quartiles and the highest quartile. Did you run the model using the lowest quartile as a reference level, and compared with other quartiles? Are they suggesting linear relationship? Or the result is indicating a threshold-model supporting your presented analysis?

9.      In Table 4, the authors showed results stratified by metropolitan or non-metropolitan. Results should be reviewed with caution, since this is a stratified result. All numbers are not directly comparable. If the authors want to discuss comparison between two regions, the authors need to run interaction model rather than stratification model.

10.  Motivation using WHO interim target-1 value is not clear. Looking Table 1, it seems PM10 WHO interim target-2 (50µg/m3) seems more making sense as a cutting point. This will be a nice to show as a sensitivity analysis.

11.   What is the circle size representing in Figure 2? Is it proportional to the population or number of the preterm birth? If it is not, same circle size should be used since color is already expressing the pollution level or prevalence of preterm.

12.  The authors discuss that pregnant women should avoid high PM10 exposure, but what kind of action can the women do? Unless they relocate or keep staying indoor with air purifier, it is hard to avoid exposure from ambient pollution. Furthermore, these two approaches may not be realistic approach for some women.

13.  Line 222-225: Last paragraph of the discussion is not relevant to the study or your primary finding. This paragraph should be developed in more detail or take out otherwise.

14.  Line 63: “air pollution would negatively influence birth outcomes, particularly preterm births and low birth weight.” Avoid using any terms implying causation. In this case, ‘influence’ should be avoided.

Author Response

Reviewer 2

Comment #1: It is not clear the way PM10 exposure was assigned to each mother. It said “We calculated monthly averages of PM10 concentrations for each address and matched it to individuals based on their gestational age.” Many previous published papers used weekly average rather than monthly to calculate gestational exposure. This will avoid exposure misclassification. If it was assigned monthly average, someone who delivered the baby on the 1st day of the month and someone who delivered the baby on the 30th in the same month will be assigned same pollution exposures as their last month of exposure. Exposure misclassification is somewhat concern; approximately 4 weeks difference of exposure period.

Comment #5: The authors evaluated two different type of birth outcomes. One is fetal growth, which is evaluated by birth weight, and the other is gestational length, which is evaluated by gestational age. In evaluating gestational length (i.e. preterm birth), exposure period should be adjusted to the relevant exposure period. Even if the exposure length is 40 weeks for a given infant, their exposure period should be adjusted to 37 weeks since exposure after 37 weeks are non-relevant to the outcome, preterm birth. This is same for very preterm birth. Any birth after 32weeks should assign pollutant level until 32nd gestational weeks

Thank you for your suggestions and we agree with your concerns. Ideally, our analysis would have been based on weekly averages. However, the Korean national birth registry provides only the month of birth and the week of gestational age for reasons of personal information protection. Given this limitation, we could not merge weekly averages and birth data. However, the data for gestational week is reliable, and we could categorize preterm and term births exactly.

Comment #2: Regarding the comment above, how much is it reliable on gestational age? Are they determined based on self-report or ultrasound? In determining gestational exposure, how did you calculate conception date?

In Korea, parents report birth information of their children at government offices. They visit the office with the medical certificate which is written by the obstetrician and report the information under the supervision of civil servants, making this information quite reliable.

Comment #3: In the limitation paragraph of the discussion, the authors wrote “there was also no data regarding the date of birth,” but the method section said date of birth is available (line 71). Please clarify this.

Thank you for your question. All parents report month and date of birth and weeks and days of gestational age, but the Korean national birth registry does not provide the date of birth and days of gestational age for reasons of privacy protection. Per your comment, we clarified these sentences in the method section (page 2, lines 74-80).

“In Korea, all parents must report their child’s birth within 1 month and include the following information: month and date of birth; maternal residential address at the time of birth; place of birth (hospital or elsewhere); parental ages; gestational age (weeks and days); sex; birth weight; birth order; total number of births; and parental education, occupation, and nationality. All data were acquired from Statistics Korea [24], which offers data (except for date of birth and days of gestational age, which are withheld for reasons of privacy protection) to all researchers who submit the objective of a study to their website.”

Comment #4: In the statistical model, the authors put infants’ sex as an adjustment variable. This should be taken out from the model. It is true that infants’ sex is strongly tied with adverse birth outcomes. Nevertheless, this variable is not associated with gestational exposure to PM10. Unless the variable is associated with exposure variable, the variable does not satisfy the requirement as an adjustment variable in the epidemiological model. Please refer to the epidemiological textbook to ensure this point.

Comment #6: In preterm analysis, the authors used birth weight as a confounder. Again, I don’t think this variable is qualified as a confounder in the epidemiological model.

Thank you for your very important comments. We chose those variables as potential confounders based on the findings from previous studies [1-3]. However, we agree with your comments that these variables likely do not qualify as confounders in the epidemiological model. We have re-analyzed our data and modified manuscript.

1) Parker, J.D.; Schoendorf, K.C.; Kiely, J.L. Associations between measures of socioeconomic status and low birth weight, small for gestational age, and premature delivery in the United States. Ann Epidemiol 1994, 4, 271-8.

2) Shin, S.H.; Lim, H-T.; Park, H-Y.; Park, S.M.; Kim, H-S. The associations of parental under-education and unemployment on the risk of preterm birth: 2003 Korean National Birth Registration database. Int J Public Health 2012, 57, 253-60.

3) Song, I.G.; Kim, M.S.; Shin, S.H.; Kim, E-K.; Kim, H-S.; Choi, S.; et al. Birth outcomes of immigrant women married to native men in the Republic of Korea: a population register-based study. BMJ Open 2017, 7, e017720.

Comment #7: Low birth weight and preterm birth is including infants delivered before 37 weeks, but birth weight is evaluated only for term infants. The number of term birth should be mentioned in the text as well as table 1.

Thank you for your comment. As preterm birth, which is a main outcome of this study, can be highly correlated with low birth weight, we analyzed PM10 concentration and birth weight only in term infants. We refer to a previous study which found that concentrations of PM2.5 during pregnancy were associated with an increased risk of low birthweight at term [1]. In order to clarify the number of term births, we added the associated calculation in the text and table 1, which provides the figure for number of term births (page 4, lines 130-131).

“Proportions of low birth weight among term infants (28,728/1,659,659 x 100) were constant at 1.7% over the observation the period.”

1) Pedersen, M.; Giorgis-Allemand, L.; Bernard, C.; Aguilera, I.; Andersen, A-M.N.; Ballester, F.; et al. Ambient air pollution and low birth weight: a European cohort study (ESCAPE). Lancet Respir Med 2013, 1, 695-704.

Comment #8: In the analysis, exposure was compared between low three quartiles and the highest quartile. Did you run the model using the lowest quartile as a reference level, and compared with other quartiles? Are they suggesting linear relationship? Or the result is indicating a threshold-model supporting your presented analysis?

Comment #10: Motivation using WHO interim target-1 value is not clear. Looking Table 1, it seems PM10 WHO interim target-2 (50µg/m3) seems more making sense as a cutting point. This will be a nice to show as a sensitivity analysis.

Thank you for your very important suggestions. At the time that we analyzed the data, we determined that the WHO interim target-2 level (50 µg/m3) was too low for a cut point, because it was very close to the mean concentration of PM10 in Korea (Table 2) during the study period. When we estimated the cut point using the Youden method, we arrived at 59.9 and 63.7 for preterm and very preterm birth, respectively, and 58.9 for low birth weight in term infants. Therefore, we chose a reference level of 70 µg/m3, which is the WHO interim target-1 level, and a well-established-long-term exposure level.

Comment #9: In Table 4, the authors showed results stratified by metropolitan or non-metropolitan. Results should be reviewed with caution, since this is a stratified result. All numbers are not directly comparable. If the authors want to discuss comparison between two regions, the authors need to run interaction model rather than stratification model.

Thank you for your comment. We agree that is important to be cautious when viewing these findings, and the readers should be aware that is a stratified analysis. We have adjusted our presentation of the findings by removing sentences from the results and discussion sections (page 7, line 170 & page 8 line 193).

Comment #11: What is the circle size representing in Figure 2? Is it proportional to the population or number of the preterm birth? If it is not, same circle size should be used since color is already expressing the pollution level or prevalence of preterm.

Thank you for your question. The circle size is proportional to either the pollution level or the prevalence of preterm birth. In ArcGIS Maps for Office, the size difference was the default method for expressing different levels, and we included two colors to represent each of the two measurements.

Comment #12: The authors discuss that pregnant women should avoid high PM10 exposure, but what kind of action can the women do? Unless they relocate or keep staying indoor with air purifier, it is hard to avoid exposure from ambient pollution. Furthermore, these two approaches may not be realistic approach for some women.

Thank you for your important suggestion. We aimed to emphasize both approaches (individual efforts and policy-making) for preventing exposure to PM10 among pregnant women. However, we understand how the recommendation for women to avoid exposure to particulate matter may appear impractical. As such, and according to your suggestion, we have modified the conclusions (page 1, line 31-32 & page 9, lines 245-250).

“Further studies are needed to explore the pathophysiologic mechanisms and guide policy development to prevent future adverse effects on birth outcomes.”

“In conclusion, we found that exposure to ambient air pollutants during pregnancy, especially PM10, was associated with an increased rate of preterm births. Future research priorities should include explorations of the pathophysiological mechanism behind this association. Long-term, multifaceted studies are also needed to guide development of policies to prevent the adverse effects of air pollutants on birth outcomes in the future.”

Comment #13: Last paragraph of the discussion is not relevant to the study or your primary finding. This paragraph should be developed in more detail or take out otherwise.

Thank you for your kind suggestion. We aimed to explain plans for future research, but decided that this is not appropriate for this article. As such, and per your suggestion, we have removed this paragraph.

Comment #14: Line 63: “air pollution would negatively influence birth outcomes, particularly preterm births and low birth weight.” Avoid using any terms implying causation. In this case, ‘influence’ should be avoided.

Thank you for your comment. We have modified the sentence (page 2, lines 67-68).

“We hypothesized that air pollution would be negatively associated with birth outcomes, particularly preterm births and low birth weight.”

Round 2

Reviewer 1 Report

the authors have improved the manuscript acording to my comments.